# Polyhydroxybutyrate as a Novel Biopolymer for Dental Restorative Materials: Biological and Morphological Analysis

**DOI:** 10.3390/polym17030313

**Published:** 2025-01-24

**Authors:** Cigdem Atalayin Ozkaya, Beliz Ertan, Gizem Kaftan Ocal, Guliz Armagan, Gokhan Gungor, Murat Demirbilek, Huseyin Tezel, Vincenzo Notaro, Nicola Scotti, Andrea Baldi

**Affiliations:** 1Department of Restorative Dentistry, School of Dentistry, Ege University, Izmir 35040, Turkey or cigdem.atalayin@ege.edu.tr (C.A.O.); belizertan26@gmail.com (B.E.); huseyin.tezel@ege.edu.tr (H.T.); 2Department of Biochemistry, Faculty of Pharmacy, Afyonkarahisar Health Sciences University, Afyonkarahisar 03030, Turkey; gizemkaftan03@gmail.com; 3Department of Biochemistry, Faculty of Pharmacy, Ege University, Izmir 35040, Turkey; guliz.armagan@ege.edu.tr; 4Innovaplast Biotechnology Inc., Eskisehir 26040, Turkey; ggungor@innovaplast.com.tr; 5Department of Biology, Polatli Faculty of Arts and Sciences, Ankara Haci Bayram Veli University, Ankara 06900, Turkey; murat.demirbilek@hbv.edu.tr; 6Department of Surgical Sciences-Prosthetic Dentistry, Dental School, University of Turin, 10129 Turin, Italy; vincenzo.notaro@unito.it (V.N.); andrea.baldi@unito.it (A.B.)

**Keywords:** biopolymer, polyhydroxybutyrate, biocompability, resin-based composite, micro-computed tomography, scanning electron microscopy

## Abstract

Polyhydroxybutyrate (PHB) is a biopolymer produced by bacteria. This study aimed to implement the production process of experimental medical-grade PHB and to evaluate its morphology and biocompatibility compared to conventional resin-based composites (RBCs). PHB raw material was produced via biological process and then the membrane was generated via electrospinning specifically for this study and imaged with Micro-Computed Tomography (Micro-CT) and scanning electron microscopy (SEM). MTS assay was used to assess the cytotoxicity of PHB compared to other materials. Test groups included two packable resin composites (Point 4-Kerr, G-aenial anterior-GC), two flowable resin composites (Filtek Ultimate Flowable-3M ESPE, Nova Compo HF-Imicryl), a compomer (Nova Compomer-Imicryl), a fissure-sealant (Fissured Nova Plus-Imicryl), and the PHB membrane (Innovaplast Biotechnology Inc., Eskisehir, Turkey). A control group consisting of cells without any test material was also produced. To perform the MTS assay, disc-shaped specimens of the aforementioned materials were prepared and then incubated with mouse fibroblast cells (L929) for 24 and 48 h. Micro-CT and SEM images revealed a homogeneous and fibrillary structure of the PHB. MTS assay revealed the highest cell viability in the PHB, Nova Compomer, and Fissured Nova Plus groups after 24 h. PHB and Nova Compomer showed the highest viability rates at 48 h while other RBCs had rates below 25% (*p* < 0.05). Considering the cell viability data and its fibrillary structure, from a biological point of view, PHB seems a promising biopolymer that might have applications in the field of dental restorative materials.

## 1. Introduction

Among the most used materials in the field of restorative dentistry, resin-based composites (RBCs) play a central role. The clinical longevity of these materials is related not only to their physical and chemical properties but also to biological aspects. As a matter of fact, modern RBCs showed in numerous studies high mechanical, physical, and aesthetic performances. However, it must be considered that these materials come into contact with dental hard tissues, soft tissues, and fluids (saliva, gingival groove fluid). Thus, biological compatibility is crucial to reduce the risks related to monomer toxicity. Regarding this matter, concerns are reported in the literature regarding the cytotoxic effects of monomeric components of commonly used RBCs [1]. The cytotoxicity pattern of RBCs varies depending on the degree of conversion, the release of free monomers, and the degradation of the resin matrix [2]. It is therefore not surprising that the research and development of more biocompatible materials are considered priority topics in modern dentistry.

Since the usage of synthetic/resin-based polymers often leads to biological drawbacks, including inadequate biocompatibility and immune response of the body, natural biopolymers have recently been applied in various fields of medicine [3,4]. Biopolymers are macromolecular compounds that are products of the vital activities of living organisms and they are obtained from biological sources, such as insects, crustaceans, and various microorganisms. Nowadays, the interest in biopolymers has increased both due to the global interest in environmental pollution and natural resources and the advantages of biological polymers over synthetic ones. This has led to the commercialization of biopolymers and related products in various fields of biomedicine, ecology, and bioengineering [5,6,7]. One of the distinguishing features of biopolymers is that their degradation by-products are not harmful to the environment and they have higher economic value and biodegradability as they are naturally derived from the ecosystem [8,9].

Among biopolymers, Polyhydroxyalkanoates (PHAs) are considered one of the most promising, with polyhydroxybutyrate (PHB) being the most well-known type [4]. PHAs have many advantages, such as biodegradability, biocompatibility, and controllable thermal and mechanical properties [10,11,12]. Moreover, oligomers and monomers released as a result of PHA biodegradation do not pose a toxic risk to cells and tissues [13]. These properties allow their use as bioimplant materials for medical and therapeutic applications [10]. Moreover, PHA is declared as an ecologically biocompatible solution for the medical and dental field considering the potential properties for tissue regeneration, development of self-healing polymers, and targeted effective treatments [14]. The PHA is reported as a convenient and green alternative for dental implant surgery [15]. PHAs can also be used to encapsulate hydrophobic drugs, providing an approach for more targeted and effective treatments. The forms of PHB (poly-3-hydroxybutyrate (P3HB), poly-4-hydroxybutyrate (P4HB), and polyhydroxyoctanoate (PHO)) have been approved by the Food and Drug Administration (FDA) [16,17]. PHA has, today, a wide range of applications, such as surgical suture material, wound dressing/healing, orthopedic applications, bone implants, scaffold material, and tissue engineering (heart, bone, cartilage, nerve, etc.) [18].

Although there are different documented usages of PHB in the field of medicine and tissue engineering, there are no studies that evaluated its potential in restorative dentistry. Thus, the aims of this in vitro study were to implement the production process of experimental medical-grade PHB and to assess (1) biocompatibility compared to conventional resin-based composites (RBCs) and (2) its morphology for further development. The main null hypothesis tested was that there are no differences between PHB and other commonly used RBCs at 24 h and 48 h in terms of cell viability.

## 2. Materials and Methods

### 2.1. Preparation of Specimens

Two resin composites (Point 4-Kerr (Santa Barbara, CA, USA), G-aenial anterior-GC (Tokyo, Japan)), two flowable resin composites (Filtek Ultimate Flowable-3M ESPE (St. Paul, MN, USA), Nova Compo HF-Imicryl (Konya, Turkey)), a compomer (Nova Compomer-Imicryl (Konya, Turkey)), a fissure sealant (Fissured Nova Plus-Imicryl (Konya, Turkey)), and an experimental membrane consisting of medical grade PHB (Innovaplast Biotechnology Inc., Eskisehir, Turkey) were tested in this in vitro study. Standardized disc-shaped specimens (6 × 2 mm) were prepared in plastic molds without polishing procedure and aseptic conditions. Regarding the RBCs included in the study, all polymerization protocols followed manufacturers’ instructions. The RBC specimens were polymerized via a 1200 mW/cm^2^ LED Curing Device (Elipar S10–3M ESPE, St. Paul, MN, USA) for 20 s from two sides and the standardized distance was set as 1 mm. On the other hand, details regarding the production process of the experimental medical-grade PHB membrane generated via electrospinning are reported in the following paragraph. A representative list of employed materials, alongside their composition, manufacturer, and lot number, is reported in Table 1.

### 2.2. PHB Production Procedure

PHB was obtained from bacteria’s fermentation, with a bacterial strain that is confidential for trading reasons. After growth, the fermented bacteria were subjected to a purification process and the experimental medical-grade PHB raw material emerged. This production process had been approved with pre-use tests (sensitization test, irritation test, hemolysis test, in vitro cytotoxicity assay, subacute systemic toxicity test, subchronic systemic toxicity test, and genotoxicity test) by the cooperated producer company (Innovaplast Biotechnology Inc., Eskisehir, Turkey). The PHB membrane production process was carried out, in accordance with the study design, by the electrospinning method. The electrospinning system consists of the power source and the injector pump. An aluminum-foil-covered copper plate was used as a collector and PHB at a concentration of 8% (*w*/*v*) was dissolved in chloroform, 2-fluoroethanol (20:1 *v*/*v*) at 60°C in a heated magnetic stirrer. PHB solution was placed in the syringe pump in a 20 mL syringe; the collector was placed 10 cm away from the syringe. The syringe pump was set to a 0.3 mL/hour flow rate, the power supply was 10 kV, and PHB membranes were produced on the copper plate [19]. The disc-shaped samples suitable for the study design were cut from the produced PHB membrane. A comprehensive scheme of the production stages of PHB is illustrated in Figure 1.

### 2.3. Morphological Analysis of PHB

The structural properties of the PHB membrane were qualitatively evaluated with 3D micro-computed tomography (Micro-CT) (Skyscan 1172-Bruker, Kontich, Belgium) and scanning electron microscopy (SEM) (Thermoscientific Phenom XL, Bleiswijk, Netherlands).

Regarding the Micro-CT process, raw data were acquired with the following parameters: voltage = 59 kV, current = 167 μA, no filter, 4 k sensor, 4 μm voxel size, averaging = 8, rotation step = 0.1°, 180° acquisition. The obtained .tif lateral projections were reconstructed with dedicated software (NRecon 1.7.4.6, Bruker, Billerica, MA, USA) into .dicom files, suitable for 3D reconstruction and analysis. The following parameters were used as a correction to improve image quality: beam hardening correction = 50%, smoothing = 2, ring artifact correction = 7. Representative images were recorded with segmentation software to highlight some aspects of the PHB 3D morphology (Mimics Medical 24.0, Materialise, Leuven, Belgium).

To obtain SEM images, a squared sample of PHB was cut and sputter-coated (100 s, 50 mA) with gold/palladium (Quorum Sputter Coater SC7620, Kent, UK). The PHB surface was then analyzed using a SEM device, to qualitatively evaluate the surface at different magnifications.

### 2.4. Biological Analysis (Cell Viability and Cytotoxicity)

The cell viability and cytotoxicity assays were performed according to ISO 10993-5 [20]. All specimens were sterilized via UV for 15 min before the cell culture process and placed into 48-well plates with one sample per well for direct contact testing. The groups were incubated with L929 mouse fibroblast cells (ATCC, CCL-1: NCTC clone 929 Aerolar Fibroblast Mouse, LOT: 70026472) for 24 and 48 h (1 × 10^4^ cell/well). The cells treated without any specimen served as a control group in the experimental model. After that MTS assay, a test method based on the colorization of viable cells, was used to assess cell viability and cytotoxicity. Following the cell culture periods, the solution containing MTS (Promega, Madison, WI, USA) and phenazine methosulfate (PMS) (Sigma Aldrich, St. Louis, MO, USA) was added to the cells for 2–3 h at 37 °C and the viable growing cells were estimated by monitoring the absorption of the product at 490 nm. The experiments were carried out in triplicate and the results were reported as the mean absorption ± standard deviation. Cell viability was calculated by comparing the absorbance of treated cells with that of the control cells and expressed as % control.

### 2.5. Statistical Analysis

GraphPad Prism software (GraphPad Prism version 7.00 for Windows, GraphPad Software, La Jolla, CA, USA) was used for statistical analysis of the cell viability data. The normality of the data distribution was determined using the Shapiro–Wilk test. A non-parametric Kruskal–Wallis test was performed followed by Dunn’s post hoc test. Differences were considered statistically significant at *p* < 0.05.

## 3. Results

### 3.1. Morphological Analysis

The structure of the obtained PHB membrane is illustrated in Figure 2 and Figure 3. The Micro-CT images show the homogeneous, compact, and reticular structure of the PHB membrane, even at high magnification (Figure 2A,B). No significant voids are noticeable alongside the whole volume.

SEM surface images highlight the fibrillar structure of the PHB membrane at different magnifications (Figure 3A–D, from 250× to 4800×). In Figure 3A (250×), it is noticeable how small precipitates/agglomerates of polymers are present but distributed in a rather uniform way inside the network. Moreover, in the same figure, it is worth mentioning that in the macrostructure of PHB, most of the fibers seem to follow a precise direction (from upper left to bottom right), probably due to the spinning technique applied for the production process. In Figure 3B (360×), the filamentous network is observable but, in this smaller scale, seems to have a more chaotic distribution of the fibers. Figure 3C (1000×) provides a better view of the aforementioned agglomerates, which have a quite constant dimension of approximately 20 microns (scale is reported in the figure for better understanding). Finally, a close-up of the fibers is reported in Figure 3D (4800×), showing that the thickness of single PHB fibers is in the scale of 2–3 microns, with spaces among fibers in the 1–5 microns range.

### 3.2. Biological Analysis

Considering the cell viability data at 24 h (Figure 4), the highest cell viability was determined in PHB, Nova Compomer, and Fissured Nova Plus groups among the tested materials (*p* < 0.05). Although the increased tendency in cell viability was observed for the PHB and Nova Compomer groups, there was no significant difference between PHB, Nova Compomer, and Fissured Nova Plus compared to the control group (*p* > 0.05). The lowest cell viability was determined in the Point 4, G-aenial anterior, Filtek Ultimate Flowable, and Nova Compo HF groups compared to the control (*p* < 0.0001).

At 48 h (Figure 5), a decrease in cell viability was observed in all groups (*p* < 0.05). The highest cell viability rate was detected in the PHB and Nova Compomer groups (*p* < 0.05). There was no significant difference between the PHB, Nova Compomer, and control groups. The lowest cell viability was determined in the Point 4, G-aenial anterior, Filtek Ultimate Flowable, Nova Compo HF, and Fissured Nova Plus groups compared to the control (*p* < 0.0001). The cell viability rate was observed as lower than 25% in these tested RBCs at 48 h.

## 4. Discussion

Biocompatibility is an essential characteristic of modern restorative materials [21]. However, despite the efforts of monomer cytotoxicity research, there is still a lack of data and room for improvements, which might also lead to the discovery of innovative materials. In this study, the production of experimental medical grade PHB was performed and the biocompatibility was evaluated comparatively with different RBCs. PHB showed to have a better biocompatible profile compared to most RBCs in terms of higher cell viability rates at 24 and 48 h, thus the null hypothesis was rejected.

The evaluation of the biocompatibility of materials constitutes a pivotal step in the process of material acceptance, in addition to the assessment of its physical properties. The cell culture studies represent the primary phase in the evaluation of biocompatibility [22]. L929 mouse fibroblast cells are recommended by international standards for the testing of medical devices and materials used in dentistry because of the easily controlled cell culture conditions [20,23] and they were also preferred in this study design. All of that, including the cell attachment properties and morphology and the interaction with the other cell types, such as human dental pulp cells and/or human osteoblast-like cells, should be investigated in further studies to clarify the usage area of PHB in the dental field.

It has been reported that commonly used RBCs generally cause moderate cytotoxic reactions in vitro at 24–72 h of contact, while cytotoxicity is significantly reduced in the presence of a dentin barrier [24]. The cytotoxic effect of RBCs is largely dependent on the light source used and the type of resin system [24]. For example, Bisphenol A-glycidyl methacrylate (Bis-GMA)-free composites are reported to have significantly lower cytotoxicity and the polishing procedure has been noticed to reduce the cytotoxicity of resin composites [25]. Moreover, an extended polymerization time has been advised to reduce the biological toxic effects of resin composites by decreasing the release of residual monomers [26]. In the present study, since there is no routine polishing procedure for the experimental material PHB, all samples were tested without a polishing procedure. On the other hand, to reduce biases related to polymerization protocol, we analyzed RBCs with a protocol (extended polymerization time to 20 s from both surfaces) intended to reduce at minimum the amount of residual monomers and maximize the degree of conversion. On the other hand, to produce the PHB, the electrospinning method was selected. This method allows the production of fiber structures from different polymers while also allowing the modification of intrinsic polymer structures and improvement of mechanical properties [27].

Concerning the biological analysis, the type of the monomer has an impact on the biocompatibility of RBCs and the main monomers, such as Bis-GMA, trimethylene glycol dimethacrylate (TEGDMA), and (urethane dimethacrylate) UDMA, have been reported to be cytotoxic [28,29]. Bis-GMA, which is the most commonly used monomer in RBCs, is also the most cytotoxic, followed by UDMA, TEGDMA, (2-hydroxyethyl methacrylate) HEMA, and methyl methacrylate (MMA) [30]. Among the materials used in this study, the cell viability decreased in the groups including these specific monomers alone or in combination. In this study, the cell viability rate at 24 h was observed as higher in the groups without these monomers, such as PHB and Fissured Nova Plus, consistent with the mentioned previous reports. Among the tested RBCs, all of them are resin composites, except the polyacid-modified resin composite Nova Compomer. Nova Compomer was declared as containing Bis-GMA in an organic matrix but the cell viability profile seems different from the other Bis-GMA-containing materials. The combination of the structure with the incorporation of glass into the inorganic matrix and prepolymer fillers appears to mitigate the release and bioactivity of monomers. This aligns with studies suggesting that the formulation and filler integration in resin-based materials can significantly influence monomer leaching and subsequent cytotoxic effects [30]. The findings underscore the importance of structural and compositional variations in determining the biocompatibility of dental materials. Specifically, the interaction between the organic matrix and inorganic fillers in compomer formulations may have altered/limited monomer release and activity in Nova Compomer. Therefore, the biocompatibility of the compomer structure should be evaluated specifically and investigated in further studies. Another remarkable finding is the decrease in the cell viability of Fissured Nova Plus at 48 h. This material did not contain the specific monomers Bis-GMA, TEGDMA, and UDMA and exhibited high cell viability at 24 h. However, the potential effect of the methacrylates that exist in its content may be manifested by a high decrease in cell viability at 48 h. Compared to the control group, the only materials with higher cell viability at both 24 h and 48 h are the PHB and Nova Compomer groups. Therefore, the findings show that PHB is the specific ‘biopolymer’ material that exhibits a high biocompatibility profile with a high cell viability rate. Cell viability refers to the number of viable cells present in a given sample while cell proliferation serves as a crucial metric for comprehending the underlying mechanisms of specific genes, proteins, and pathways involved in cell survival or death subsequent to exposure to toxic substances [31]. In this study, the incubation period was performed as 24 and 48 h. Although cell damage may occur, it may not result in death at an early culture period, such as 24 h. Therefore, we applied two different time points, 24 and 48 h. Various tests are currently used to determine the biocompatibility of a material. The evaluation of biological properties of materials usually starts with simple in vitro test methods using cell cultures and continues with animal tests. The biocompatibility with animal models may be designed with longer periods to determine the biocompatibility profile of the material in the long term.

Regarding the morphological aspects, PHB obtained from the electrospinning technique seems to be promising for further development. In fact, as mentioned before, this method is flexible and it allows the modification of the polymer structure by changing some parameters, ultimately acting on its mechanical properties [27]. With the setting that was applied in the present study, the obtained membrane was uniform, compact, and with a neat macro-structure, as shown in Figure 3 and Figure 4. This homogeneous aspect is an essential characteristic of any restorative dental materials. Moreover, the thickness of the PHB fibers (two to three microns) and the voids in between the fibers (one to five microns) could perfectly allow the infiltration of both resin and ceramic particles. This could lead to the development of novel composite materials, with good mechanical properties and high biocompatibility. Moreover, the surface hydrophilicity, surface free energy, and biodegradability of PHB may be considered interesting properties for tissue–biomaterial interactions. However, chronic toxicity, the effects of material degradation products, and the effects on other cell types should be evaluated in further studies to specificize the usage point of PHB in dentistry.

Given these preliminary tests, more comprehensive studies should be carried out. The biological profile of the material should be completely identified by antibacterial activity and inflammatory response analyses. Moreover, the mechanical properties of the material, such as microhardness, adhesive strength to tooth structures and restorative materials, water sorption, water solubility, microleakage, shrinkage, shear strength, tensile strength, surface morphology, biodegradation process, and resistance to intraoral conditions, should be investigated in further studies to identify the specific application fields of PHB as a dental biomaterial. However, within the limitation of the present in vitro preliminary study, it can be concluded that PHB seems a promising biocompatible biopolymer that might have applications in the field of dental restorative materials. For example, considering the wide range of applications of PHB, such as wound dressing/healing, scaffold material, and tissue engineering, it could be also used for pulp-capping and repair in the dental field, due to the high biocompatibility profile, and potentially pave the way for applications in tissue engineering.

## Figures and Tables

**Figure 1 polymers-17-00313-f001:**
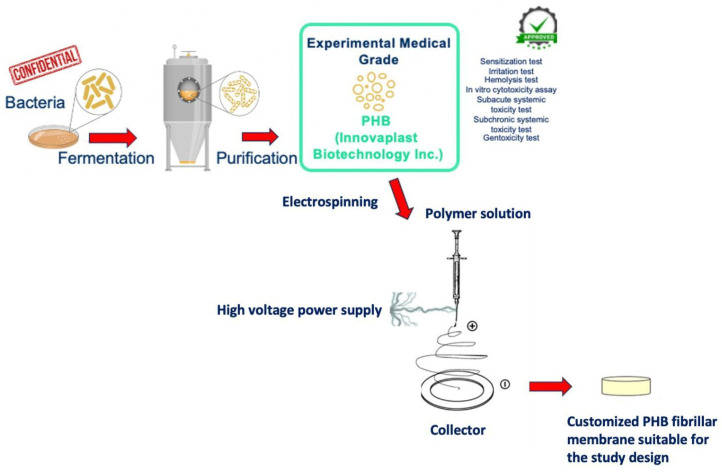
A schematic, step-by-step illustration of medical-grade PHB membrane production stages.

**Figure 2 polymers-17-00313-f002:**
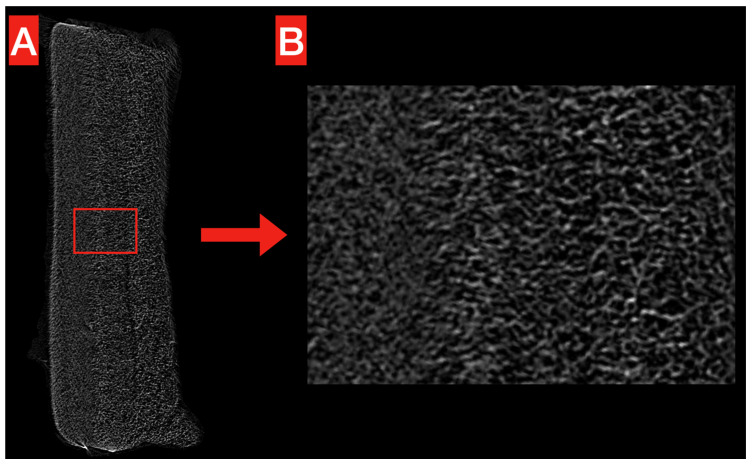
Micro-CT exemplificative image of the obtained PHB membrane. The overall structure (**A**) looks homogeneous and compact, without significant voids. This is confirmed at higher magnification (**B**), where the reticular structure looks more clear.

**Figure 3 polymers-17-00313-f003:**
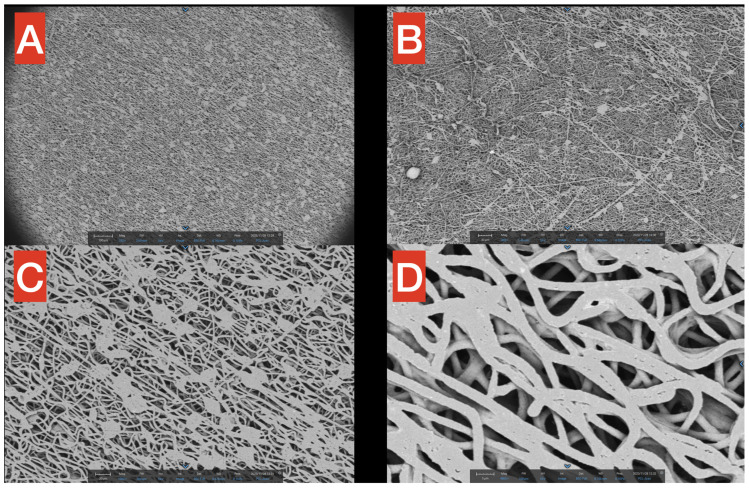
SEM images of the PHB membrane at different magnifications (**A**–**D**), from 250× to 4800×. The macrostructure of the PHB is filamentous and uniform, with the exception of small aggregates at higher magnifications.

**Figure 4 polymers-17-00313-f004:**
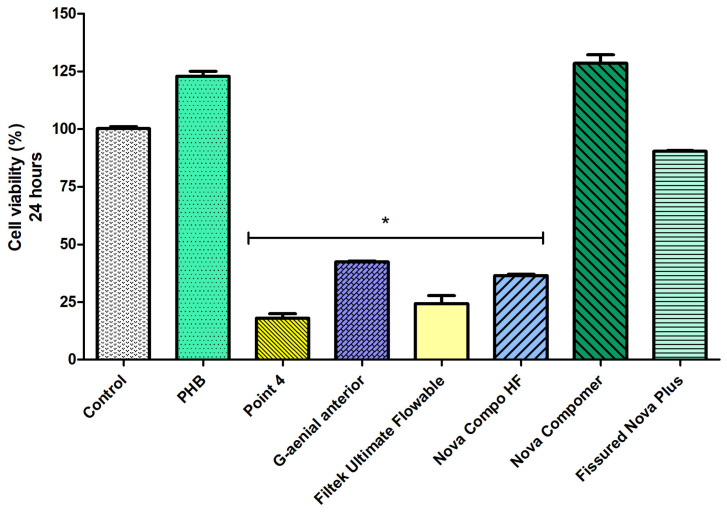
The cell viability rates at 24 h (* significant difference compared to the control group, *p* < 0.0001).

**Figure 5 polymers-17-00313-f005:**
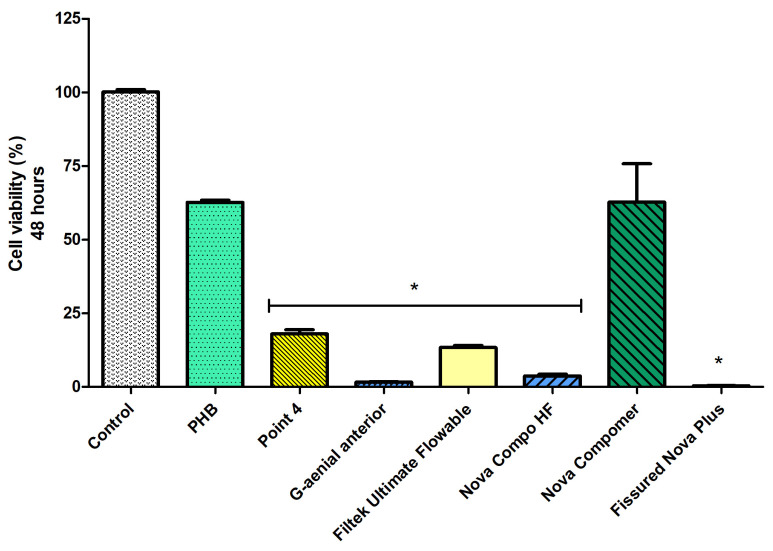
The cell viability rates at 48 h (* significant difference compared to the control group, *p* < 0.0001).

**Table 1 polymers-17-00313-t001:** Composition, manufacturer, and lot number of tested materials. Details are listed as declared by manufacturers.

Material	Composition	Manufacturer	Lot Number
Point 4	BisGMA, TEGDMA, BisEMA, Barium aluminum boro silicate, filler 77% by weight	Kerr, Santa Barbara, CA, USA	7500391
G-aenial Anterior	UDMA, dimethacrylate co-monomers, prepolymerized silica, strontium fluoride	GC, Tokyo, Japan	2007021
Filtek Ultimate Flowable	The Resin Matrix: BIS-GMA, TEGDMA, EDMA, benzotriazole, diphenyl iodonium hexafluorophosphate, dimethacrylate, ytterbium flüoride. The Filler: 65 wt %, 46 vol % Silica (75 nm), zirconium (5–10 nm), silane treated ceramic, silica.	3M ESPE, St. Paul, MN, USA	NA83839
Nova Compo HF	Organic Matrix Content: Hydrophobic aromatic dimethacrylates, Bis-GMA, Bis-MEP, TEGDMA, UDMA. Inorganic Filler Particles: Silanized barium glass, nano ytterbium, silanized highly dispersed nano silicon dioxide, silica-zirconia, prepolymer fillers 65–70% by weight, 53–55% by volume	Imicryl, Konya, Turkey	1783
Nova Compomer	Organic Matrix Content: BIS-GMA, Dimethacrylate ULS. Inorganic Matrix Content: Barium Glasses, Ytterbium trifluoride, Prepolymer filler by Weight: 78%, by Volume: 59–60	Imicryl, Konya, Turkey	21D738
Fissured Nova Plus	Hydrophilic Dimethacrylates, hydrophobic dimethacrylates, highly dispersed silica, sodium fluoride, fluorosilicate glass, stabilizers, catalysts. Filler ratio: 55%.	Imicryl, Konya, Turkey	21C574

## Data Availability

Derived data supporting the findings of this study are available from the corresponding author [N.S.] upon eligible request.

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
