# Peer review of "Polyhydroxybutyrate as a Novel Biopolymer for Dental Restorative Materials: Biological and Morphological Analysis"

_polymers, 2025, doi:10.3390/polym17030313_

Round 1

Reviewer 1 Report

Comments and Suggestions for Authors

Comments to the Author:

This manuscript (polymers-3379205) entitled “Polyhydroxybutyrate as a Potential Biopolymer Alternative to 2 Dental Resin Materials: A Preliminary Study” by Nicola Scotti et al. reports on the production process of polyhydroxybutyrate (PHB) as a medical biopolymer and its biocompatibility when compared with various resin-based composites in dentistry. The findings suggest that PHB is a highly compatible alternative for dental applications.

According to the following weaknesses, I don't think the contribution of this manuscript is significant and deserves to be published in the Journal of Polymers in the present form. Major revision recommended for publication of the manuscript.

The following consideration has the objective to assist authors in improving their work:

The material mentioned by the authors for comparison is intended for dental 3D printing, which requires high precision and accuracy for applications such as splints, model resins, and other dental devices. These resins are polymerized using advanced 3D printing technologies like SLA, DLP, and LCD. However, comparing PHB with those resin-based products is inappropriate in this context. Instead, the authors should compare PHB with similar products like PHB.

While the electrospinning technique used to produce PHB is detailed, the choice of solvents, such as chloroform and 2-fluoroethanol, raises concerns about potential residual toxicity in the final PHB product. It is unclear whether the membranes were tested for residual solvents, and this omission leaves questions about their safety and biocompatibility unanswered.

The statistical methodology employed in the study, including the use of Kruskal-Wallis and Dunn’s post-hoc tests, is adequately described. However, the presentation of data in graphs lacks detailed statistical annotations, such as p-values between groups, which undermines the robustness and clarity of the results.

Additionally, the bacterial strain used for PHB production is noted to be confidential, which is understandable for commercial reasons. However, this lack of transparency prevents reproducibility by other researchers, a fundamental aspect of scientific rigor and validity.

The study evaluates cell viability only at 24 and 48 hours, which is insufficient to assess the long-term biocompatibility of PHB. For example, materials used for dental splints, which are classified as Class II medical devices, require longer-term evaluation to comply with FDA guidelines based on ISO 10993. Chronic toxicity and the effects of material degradation products should be assessed over weeks or months to ensure safety in clinical applications.

Lastly, the claim of a "122% increase in cell viability" for PHB lacks clarity regarding the baseline reference for this comparison. It is essential to specify whether this increase is relative to the control group or other RBC materials to improve the interpretability and validity of the results.

And so on ….

It is my feeling that this paper, in its current state, is not of sufficient quality to be published in the Journal of Polymers. It requires major rewriting before the authors consider submission again.

Author Response

Authors would like to thank the reviewers for their precious time and efforts in giving advice to improve the paper. A whole English revision was carried out in the text. Results and discussions were improved, adding a specific section for the morphological analysis that was performed. Please do not hesitate to contact us for further details and clarifications!

REVIEWER 1

The material mentioned by the authors for comparison is intended for dental 3D printing, which requires high precision and accuracy for applications such as splints, model resins, and other dental devices. These resins are polymerized using advanced 3D printing technologies like SLA, DLP, and LCD. However, comparing PHB with those resin-based products is inappropriate in this context. Instead, the authors should compare PHB with similar products like PHB.

Authors agree with this observation, text was revised accordingly. PHB alone does not possess the properties necessary to be compared with such materials, thus this reference was removed.

While the electrospinning technique used to produce PHB is detailed, the choice of solvents, such as chloroform and 2-fluoroethanol, raises concerns about potential residual toxicity in the final PHB product. It is unclear whether the membranes were tested for residual solvents, and this omission leaves questions about their safety and biocompatibility unanswered.

Pre-use tests were carried out by the cooperated producer company (sensitization test, irritation test, hemolysis test, in vitro cytotoxicity assay, subacute systemic toxicity test, subchronic systemic toxicity test and gentoxicity test). This part was clarified in the text.

The statistical methodology employed in the study, including the use of Kruskal-Wallis and Dunn’s post-hoc tests, is adequately described. However, the presentation of data in graphs lacks detailed statistical annotations, such as p-values between groups, which undermines the robustness and clarity of the results.

The results section was revised by adding the p values  for comparing the experimental groups  with control. The graphics were revised with the superscripts indicating statistical significance

Additionally, the bacterial strain used for PHB production is noted to be confidential, which is understandable for commercial reasons. However, this lack of transparency prevents reproducibility by other researchers, a fundamental aspect of scientific rigor and validity.

 PHB was biosynthesized utilizing a halophilic bacteria. Unfortunately, due to commercial reasons we are not able to provide further details about the bacterial strain. However, further studies could benefy from using the same manufacturer, that we reported in the text, for data consistency.

The study evaluates cell viability only at 24 and 48 hours, which is insufficient to assess the long-term biocompatibility of PHB. For example, materials used for dental splints, which are classified as Class II medical devices, require longer-term evaluation to comply with FDA guidelines based on ISO 10993. Chronic toxicity and the effects of material degradation products should be assessed over weeks or months to ensure safety in clinical applications.

 Authors agree with the comment. This study was a preliminary evaluation that was meant to understand if further studies and efforts are legit. Due to promising results, we will surely carry out these tests, as well as other tests for mechanical aspects, in further studies. This aspect was inserted in the limitations of the study in the discussion section.

Lastly, the claim of a "122% increase in cell viability" for PHB lacks clarity regarding the baseline reference for this comparison. It is essential to specify whether this increase is relative to the control group or other RBC materials to improve the interpretability and validity of the results.

This rate had been declared in the text to mention the non-cytotoxic effect of PHB considering cell viability rate at 24 hours. The result section was revised as follows ‘Although, the increase tendency in cell viability was observed for PHB and Nova Compomer groups, there was no-significant difference between PHB, Nova Compomer and Fissured Nova Plus compared to control group (p>0.05).’ to clarify the issue by statistical analyses.

Reviewer 2 Report

Comments and Suggestions for Authors

Introduction, improve references, 

Antibacterial effect of polymethyl methacrylate resin base containing TiO2 nanoparticles. Coatings. 2022, vol. 12, no. 11, str. 1-17.

Other characterizations needs to be performed.

SEM micropragphs shoud be included

Iprove discussion

Comments on the Quality of English Language

should be improved

Author Response

Introduction, improve references

Authors thanks for the comment, introduction was rewritten to clarify the scientific background and improve references, according to reviewer suggestions.

Other characterizations needs to be performed.

 Authors agree with the comment. This study was a preliminary evaluation that was meant to understand if further studies and efforts are legit. Due to promising results, we will surely carry out these tests, as well as other tests for mechanical aspects, in further studies. This aspect was inserted in the limitations of the study in the discussion section.

SEM micropragphs shoud be included

Authors thanks for the comment, SEM images were added in improved quality. Also, a specific paragraph for morphological aspects was created in all sections.

Improve discussion

Discussion was completely revised to make it more fluent and clear.

English language should be improved

All the manuscript was revised and checked by an English copywriter.

Reviewer 3 Report

Comments and Suggestions for Authors

The manuscript by Ozkaya and co-authors evaluated PHB and different resin-based composites regarding their compatibility with mouse fibroblast cells.

Major comments

1. The authors should consider running more experiments to strengthen the study, for example, studying the cell attachment morphology by SEM; and the cytotoxicity to other cell types, such as Human dental pulp cells (HDPCs) or Human osteoblast-like cells (MG-63).

Minor comments

L 26-40. The abstract is not clear and should be rewritten.

L 32. 'PHB was produced via electrospinning' is not correct. According to the methods section, PHB was produced via a biological process and electrospinning was used to prepare the fibrillar membrane. Please rewrite.

L 78-79. Recent manuscripts discussed the use of PHA for dental applications. Please check and cite if considered adequate - https://doi.org/10.3390/ma17225415 ; 10.18063/ijb.v9i2.655 for example.

L 80. 'An experimental medical grade PHB generated' - please rewrite for clarity. PHB specimen? PHB implant?

L 125. Please clarify the choice of L929 mouse fibroblast cells.

L 134-135. Was a DMSO treatment used as a positive control? Please consider adding this as a control.

L 143. Figure 2 is not informative. Please clarify its contribution to the manuscript.

L 163-166. Graph 1 and Graph 2 should be renamed as Figure 1 and Figure 2, accordingly.

L 186-187. Please discuss the possible reasons why the cell viability increased. Were the composites used as carbon sources?

L 247. It is unclear why PHB was considered 'the sole polymer material exhibiting a consistently high biocompatibility profile' since the Nova compomer has a similar profile (Graph 1 and 2).

Comments on the Quality of English Language

Some portions of the text must rewritten.

Author Response

The authors should consider running more experiments to strengthen the study, for example, studying the cell attachment morphology by SEM; and the cytotoxicity to other cell types, such as Human dental pulp cells (HDPCs) or Human osteoblast-like cells (MG-63).

Authors agree with the comment. This study was a preliminary evaluation that was meant to understand if further studies and efforts are legit. Due to promising results, we will surely carry out these tests, as well as other tests for mechanical aspects, in further studies. This aspect was inserted in the limitations of the study in the discussion section.

L 26-40. The abstract is not clear and should be rewritten.

Abstract was rewritten for clarification and to make it more fluent.

L 32. 'PHB was produced via electrospinning' is not correct. According to the methods section, PHB was produced via a biological process and electrospinning was used to prepare the fibrillar membrane. Please rewrite.

Authors agree with the comment, text was revised accordingly to distinguish the two steps.

L 78-79. Recent manuscripts discussed the use of PHA for dental applications. Please check and cite if considered adequate - https://doi.org/10.3390/ma17225415; 10.18063/ijb.v9i2.655 for example.

Thank you for your suggestion. The related references and contents were added Authors thanks for the comment, a revision was carried out to include these references into the text. (L79-83).

L 80. 'An experimental medical grade PHB generated' - please rewrite for clarity. PHB specimen? PHB implant?

This part was rewritten to specify the form of the PHB (membrane).

L 125. Please clarify the choice of L929 mouse fibroblast cells.

In the method section, it was clarified as ‘The cell viability and cytotoxicity assays were performed according to ISO 10993-5 (International Organization for Standardization, ISO 10993-5.Biological evaluation of medical devices-part 5. Tests for cytotoxicity: in vitro methods. Geneve: ISO, 1992.) Moreover, this issue was mentioned in the discussion section as follows: The evaluation of the biocompatibility of materials constitutes a pivotal step in the process of material acceptance, in addition to the assessment of its physical properties. The cell culture studies represent the primary phase in the evaluation of biocompatibility (Polyzois GL, Hensten-Pettersen A, Kullmann A. An assess-ment of the physical properties and biocompatibility of threesilicone elastomers. J Prosthet Dent 1994;71:500–504). L929 mouse fibroblast cells are recommended by international standards for the testing of medical devices and materials used in dentistry because of the easily controlled cell culture conditions (International Organization for Standardization, ISO 10993-5.Biological evaluation of medical devices-part 5. Tests for cytotoxicity: in vitro methods. Geneve: ISO, 1992. International Organization for Standardization, ISO 7405.Dentistry-Preclinical evaluation of biocompatibility of medicaldevices used in dentistry-Test methods for dental materials.Geneve: ISO, 1997.) For all that, the cell attachment properties and morphology, the interaction with the other cell types, such as human dental pulp cells and/or human osteoblast-like cells should be investigated in further studies to clarify the ​​usage area of PHB in dental field.

L 134-135. Was a DMSO treatment used as a positive control? Please consider adding this as a control.

The direct contact test was applied in this study design and DMSO was not used as an extraction solution. Thus, the cells were not exposed to DMSO (0.1%) as a vehicle control group.

L 143. Figure 2 is not informative. Please clarify its contribution to the manuscript.

Figures 2 and 3 were completely revised, adding a comprehensive analysis and discussion of what it is possible to see.

L 163-166. Graph 1 and Graph 2 should be renamed as Figure 1 and Figure 2, accordingly.

Figure order and naming was revised

L 186-187. Please discuss the possible reasons why the cell viability increased. Were the composites used as carbon sources?

The increase in cell viability had been declared in the text to mention the non-cytotoxic effect of PHB. The result section was revised as follows ‘Although, the increase tendency in cell viability was observed for PHB and Nova Compomer groups, there was no-significant difference between PHB, Nova Compomer and Fissured Nova Plus compared to control group (p>0.05).’ to clarify the issue by statistical analyses.

This study was designed to evaluate the biocompatibility of materials by direct contact test method. Any comparison group and analysis process for the evaluation of the usage of test materials as carbon sources and their effects on cell viability were not included. Therefore, additional analysis and data are needed to discuss this issue.

L 247. It is unclear why PHB was considered 'the sole polymer material exhibiting a consistently high biocompatibility profile' since the Nova compomer has a similar profile (Graph 1 and 2).

In this study different resin based composites (RBC) and an experimental biopolymer ‘PHB’ were tested. Among the tested RBC, all of them are resin composites except the polyacid modified resin composite Nova Compomer. Nova Compomer was declared as containing organic matrix, inorganic matrix with the incorporation of glass and prepolymer fillers.  This modified structure may have altered/limited monomer release and activity in Nova Compomer compared to the other RBC and may have contributed to the results as high cell viability rate both at 24 and 48 hours. On the other hand, PHB seem as specific ‘biopolymer’ material that exhibits a high biocompatibility profile with the high cell viability rate in both time intervals. This situation was explained in the discussion part.

Round 2

Reviewer 1 Report

Comments and Suggestions for Authors

Comments to the Author:

The authors have made sufficient changes and addressed most of my concerns. The revised manuscript can be published in the journal.

Author Response

Thanks for the comments and the precious time you have spent to improve our manuscript.

Reviewer 2 Report

Comments and Suggestions for Authors

How is with the cell viability after one week?

Fig 3. caption should be improved.

There is no application!

No conclusions!

Comments on the Quality of English Language

can be improved

Author Response

How is with the cell viability after one week?

Thank you for your suggestion.

The cell viability and cytotoxicity assays were performed according to ISO 10993-5 (International Organization for Standardization, ISO 10993-5.Biological evaluation of medical devices-part 5. Tests for cytotoxicity: in vitro methods. Geneve: ISO, 1992.). The standard ISO 10993-5 defines an incubation time of at least 24 h or, if necessary, longer until the cells are subconfluent. In this study, the incubation period was performed as 24 and 48 hours. Although the cell damage may occur, it may not result in death at early culture period as 24 hours. Therefore, we applied two different time points as 24 and 48 hours.  This situation was also explained in the discussion section as ‘In this study, the incubation period was performed as 24 and 48 hours. Although the cell damage may occur, it may not result in death at early culture period as 24 hours. Therefore, we applied two different time points as 24 and 48 hours.  The various tests are currently used to determine the biocompatibility of a material. The evaluation of biological properties of materials usually starts with simple in vitro test methods using cell cultures and continues with animal tests. The biocompability with animal models may be designed with longer periods to determine the biocompability profile of the material in long term.’.

The mentioned cell culture period as one week may also exceed the passage time of L929 cells with high proliferation rate. Therefore, it may be possible to determine the cell viability/biocompability in long term with animal-model studies.

Fig 3. caption should be improved.

  • Thanks for the comment. Fig.3 caption was improved accordingly.

There is no application!

  • Authors thanks for the comment.

PHB is used in a number of fields, including the production of sutures, stents, wound dressings, cardiovascular dressings, heart valves, orthopaedic pins, articular cartilage and adhesion barriers. However dental applications of PHB should be developed. Indeed, in the present membrane form, the material has no direct applications in dental field. But, the usage of PHB as a wound dressing in the medical field is suggested that it could also be used as a pulp capping material for the exposed pulp, which could be considered as a wound surface in the field of dentistry.

However, as explained in the discussion section, this is a preliminary study and further tests and formulation will be evaluated. A possible application that we are testing is the usage as pulp-capping material, due to high biocompatibility. This aspect was underlined in the end of discussion.

No conclusions!

  • Conclusion paragraph was removed as requested by previous editor/reviewers and approved in the current form.

Reviewer 3 Report

Comments and Suggestions for Authors

I thank the authors for considering and replying to all the comments.

Please correct the text (e.g. L27, L38, and others) to clarify that PHB was produced via a biological process and that the PHB specimens/membranes/devices, whichever fits best, were generated/produced using electrospinning. This is not clear and should be corrected.

Author Response

I thank the authors for considering and replying to all the comments.

Please correct the text (e.g. L27, L38, and others) to clarify that PHB was produced via a biological process and that the PHB specimens/membranes/devices, whichever fits best, were generated/produced using electrospinning. This is not clear and should be corrected.

Thanks for the comments and the precious time you have spent to improve our manuscript.

In this study design, firstfull medical grade PHB raw material was biosynthesized utilizing halophilic bacteria and then PHB membrane production process was carried out by the electrospinning method. This was explained in the method section in detail. The related terms were also revised in the abstract and text to clarify this aspect.

Round 3

Reviewer 2 Report

Comments and Suggestions for Authors

accept